# NIR-Assisted Image Denoising: A Selective Fusion Approach and A Real-World Benchmark Dataset

## Abstract

Despite the significant progress in image denoising, it is still challenging to restore fine-scale details while removing noise, especially in extremely low-light environments. Leveraging near-infrared (NIR) images to assist visible RGB image denoising shows the potential to address this issue, becoming a promising technology. Nonetheless, existing works still struggle with taking advantage of NIR information effectively for real-world image denoising, due to the content inconsistency between NIR-RGB images and the scarcity of real-world paired datasets. To alleviate the problem, we first propose an efficient Selective Fusion Module (SFM), which can be plug-and-played into the advanced denoising networks to merge the deep NIR-RGB features. Specifically, we sequentially perform the global and local modulation for NIR and RGB features, and then integrate the two modulated features. Furthermore, we present a real-world NIR-Assisted Image Denoising (NAID) dataset, which covers diverse scenarios as well as various noise levels and is expected to serve as a benchmark for future research. Extensive experiments on both synthetic and our real-world datasets demonstrate that the proposed method achieves better results than state-of-the-art ones. The dataset, codes, and pre-trained models will be publicly available.

## 1 Introduction

In low-light conditions, it's common to use short exposure time and high ISO in imaging to prevent motion blur, while this approach inevitably introduces noise due to the limited number of photons captured by camera. With the development of deep learning (He et al., 2016; Liang et al., 2021; Vaswani et al., 2017), many image denoising methods (Zhang et al., 2017; 2018a; Abdelhamed et al., 2020; Zamir et al., 2022; Wang et al., 2022; Zhang et al., 2022; Li et al., 2023) have been proposed to remove the noise. Although great progress has been achieved, it is still challenging for these methods to recover fine-scale details faithfully due to the severely ill-posed nature of denoising. A practical solution is burst denoising (Mildenhall et al., 2018; Godard et al., 2018; Pearl et al., 2022; Wu et al., 2023), in which multiple successive frames are merged to improve performance. But it is susceptible to the misalignment between frames, and may be less effective in facing dynamic scenes.

Fortunately, near-infrared (NIR) images with low noise can be captured at a cheap cost and utilized to enhance the denoising of visible RGB images, which has attracted increasing attention (Lv et al., 2020; Wu et al., 2020; Jin et al., 2022; Wan et al., 2022). Specifically, on the one hand, the NIR band lies outside the range of the human visible spectrum. It enables us to turn on an NIR light that is imperceptible to humans, thus capturing NIR images (Fredembach & Süsstrunk, 2008) with a low noise level. On the other hand, modern CMOS sensor is sensitive to partial near-infrared wavelengths (Xiong et al., 2021), thus allowing NIR signals to be acquired cheaply and conveniently.

Nevertheless, the inconsistencies between NIR and RGB content limit the positive effect of NIR images in denoising. Firstly, NIR images are captured under additional NIR light and are monochromatic, which leads to brightness and color discrepancies between the two modalities. Secondly, the NIR images may 'more-see' or 'less-see' the objects than the visible light ones, primarily due to inherent differences in the optical properties within each spectral domain (Fredembach & Süsstrunk, 2008). For example, as shown in Fig. 1 (a), the RGB image clearly contains textual information,

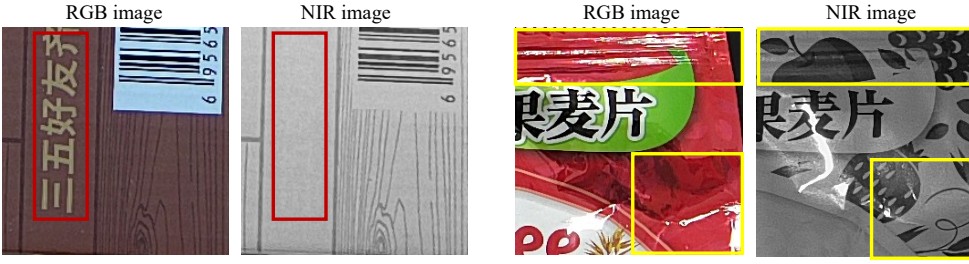

(a) Structure visible in RGB image but not in NIR image.  (b) Structure visible in NIR image but not in RGB image.

Figure 1: Examples of the structure discrepancy between RGB-NIR images. (**a**) The structure is visible in the RGB image but not in the NIR image, as shown in the red box. (**b**) The structure is visible in the NIR image but not in the RGB image, as shown in the yellow box.

while the corresponding NIR image lacks that. In Fig. 1 (b), the NIR image exhibits extra fruit patterns, while these patterns are absent in the RGB image. DVD (Jin et al., 2022) and SANet (Sheng et al., 2023) have noticed this problem, but their solutions are both complex and less effective. Additionally, due to the lack of real-world paired datasets, existing methods mainly focus on processing synthetic noisy images. NIR-assisted real-world noise removal is rarely explored.

In this work, on the one hand, we focus on dealing with the content inconsistency problem, and hope to construct a simple yet effective RGB-NIR fusion module that can be easily integrated into the existing denoising networks. Specifically, we propose a lightweight Selective Fusion Module (SFM), which consists of a Global Modulation Module (GMM), a Local Modulation Module (LMM), and a fusion operation. GMM and LMM mainly handle color and structure inconsistency issues, respectively. They predict and assign soft weights to NIR and RGB features, thus preparing for subsequent feature fusion. On the other hand, we introduce NIR-Assisted Image Denoising (NAID) dataset for NIR-assisted real-world noise removal. It encompasses diverse scenarios and various noise levels, providing a valuable resource for evaluating and promoting research in this field. We conduct extensive experiments on both synthetic DVD (Jin et al., 2022) and our real-world NAID datasets. The results show that the proposed method performs better than state-of-the-art ones.

Our contributions can be summarized as follows:

(1) For NIR-assisted image denoising, we propose a plug-and-play selective fusion module to handle content inconsistency issues between RGB-NIR images, which assigns appropriate fusion weights to the deep NIR and RGB features by global and local modulation modules.

(2) We construct a paired NIR-assisted real-world image denoising dataset with diverse scenarios and various noise levels, which has the potential to promote further research in this field.

(3) Extensive experiments on both synthetic and our real-world datasets demonstrate that our method achieves better results than state-of-the-art ones.

## 2 RELATED WORK

### 2.1 SINGLE IMAGE DENOISING

With the advancements in deep learning (Ronneberger et al., 2015; He et al., 2016; Vaswani et al., 2017), numerous single-image denoising methods (Zhang et al., 2017; 2018a; Abdelhamed et al., 2020; Chen et al., 2021; Zamir et al., 2022; Wang et al., 2022; Zhang et al., 2022; Li et al., 2023) have emerged. DnCNN (Zhang et al., 2017) pioneers the utilization of deep learning techniques and surpasses traditional patch-based methods (Buades et al., 2005; Dabov et al., 2007; Gu et al., 2014) on Gaussian noise removal. Recently, some methods (Zamir et al., 2021; Wang et al., 2022; Zamir et al., 2022; Chen et al., 2022) are developed with advanced architectures. For example, MRPNet (Zamir et al., 2021) applies a multi-stage architecture for progressive image restoration and achieves remarkable performance. Uformer (Wang et al., 2022) introduces the locally-enhanced transformer by employing the non-overlapping window-based self-attention. Restormer (Zamir et al., 2022) further reduces the computation cost by modifying the self-attention calculation from the spatial dimension to channel one. NAFNet (Chen et al., 2022) proposes a simple baseline that does not apply

nonlinear activation. Despite the significant progress achieved by these methods, the performance is still unsatisfactory when handling images with high-level noise captured under low-light conditions.

## 2.2 NIR-ASSITED IMAGE RESTORATION

Compared to single-image restoration, NIR images have the potential to assist in restoring details from degraded images. The earlier work (Krishnan & Fergus, 2009) utilizes gradient constraints for NIR-assisted image denoising. Wang *et al.* (Wang et al., 2019b) further improves the performance with deep learning methods. SSN (Wu et al., 2020) proposes a multi-task deep network with state synchronization modules. TC-GAN (Yang et al., 2021) fuses the NIR images and RGB ones based on a texture conditional generative adversarial network. DCMAN (Cheng et al., 2023) employs spatial-temporal-spectral priors to introduce NIR videos for low-light RGB video restoration. However, these methods have overlooked the color and structure inconsistency issues between the NIR images and RGB ones. CCDFuse (Zhao et al., 2023) addresses this issue by combining the local modeling ability of convolutional blocks and the non-local modeling ability of transformer ones to extract local and global features of NIR and RGB images respectively. SANet (Sheng et al., 2023) proposes a guided denoising framework by estimating a clean structure map for the noisy RGB image. Wan *et al.* (Wan et al., 2022) disentangle the color and structure components from the NIR images and RGB ones. Besides, a few works (Deng & Dragotti, 2020; Xu et al., 2022b; Jin et al., 2022) incorporate different priors into the network design, like sparse coding (Deng & Dragotti, 2020), deep implicit prior (Xu et al., 2022b) and deep inconsistency prior (Jin et al., 2022). However, their complexity makes it difficult to be integrated into existing advanced restoration networks, hindering their extensions and improvements.

## 2.3 DATASETS FOR NIR-ASSISTED IMAGE RESTORATION

Existing available NIR-RGB datasets suffer from limitations such as scarcity of data samples (Krishnan & Fergus, 2009), absence of paired real-world RGB noisy images (Brown & Süsstrunk, 2011; Zhi et al., 2018; Jin et al., 2022; Lv et al., 2020), or lack of public accessibility (Wang et al., 2019a; Lv et al., 2020). For instance, Krishnan *et al.* (Krishnan & Fergus, 2009) develop a prototype camera to capture image pairs under varying low-light conditions but only containing 5 image pairs. Its size is too small to fulfill the demands of data-driven deep-learning algorithms. IVRG (Brown & Süsstrunk, 2011) and RGB-NIR Srereo (Zhi et al., 2018) construct datasets consisting of RGB and NIR image pairs for image recognition and stereo matching, respectively. DVD (Jin et al., 2022) captures images within a controlled light-box environment. However, these datasets comprise solely clean RGB and NIR image pairs, lacking real-world noisy RGB images. Burst Dataset (Wang et al., 2019a) captures real-world noisy images by a mobile imaging device that is sensitive to both near-infrared and near-ultraviolet signals. Lv *et al.* (Lv et al., 2020) introduces the VIS-NIR-MIX dataset which utilizes a motorized rotator to manipulate illumination conditions. But they are not publicly available. The scarcity of real-world datasets has hampered future research. To address this limitation, we introduce the NIR-Assisted Image Denoising (NAID) benchmark dataset, which encompasses diverse scenarios and various noise levels.

## 3 REAL-WORLD NIR-ASSISTED IMAGE DENOISING DATASET

Existing publicly available NIR-assisted image denoising datasets generally lack real-world noisy RGB images paired with the clean RGB and NIR images, which limits the investigation in real-world NIR-assisted image denoising. To break such a limitation, we build a NIR-Assisted Image Denoising (NAID) dataset. Specifically, we employ high ISO and short exposure time to capture the real-world noisy RGB images, as shown in Fig. 2 (a). The camera's ISO and exposure time are adjusted to capture images with different noise levels. For capturing the corresponding clean RGB images, we lower the ISO of the camera and appropriately increase the exposure time, as shown in Fig. 2 (b). To obtain paired NIR images, we activate NIR light to ensure a sufficient supply of NIR illumination and then capture the NIR ones with a dedicated NIR camera, as shown in Fig. 2 (c).

All images are captured with the Huawei X2381-VG camera, which is equipped with a built-in NIR illuminator specifically designed for capturing NIR images. To ensure image registration among multiple captures, we securely position the camera and develop a remote control application to

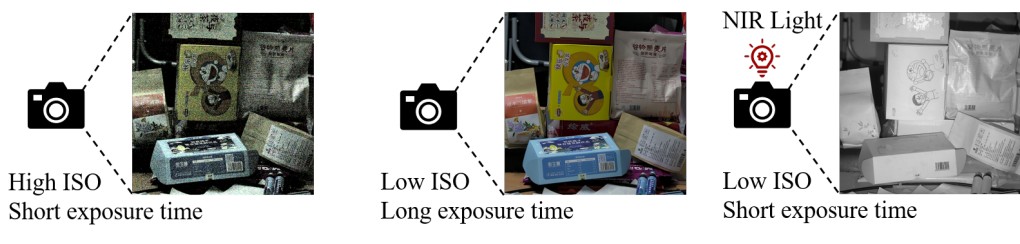

(a) Capture noisy RGB images. (b) Capture clean RGB images. (c) Capture clean NIR images.

Figure 2: The construction of NIR-Assisted Image Denoising (NAID) dataset. (**a**) Capture noisy RGB images with high ISO and short exposure time. (**b**) Capture clean RGB images with low ISO and long exposure time. (**c**) Turn on the NIR light, then capture the clean NIR images with low ISO and short exposure time.

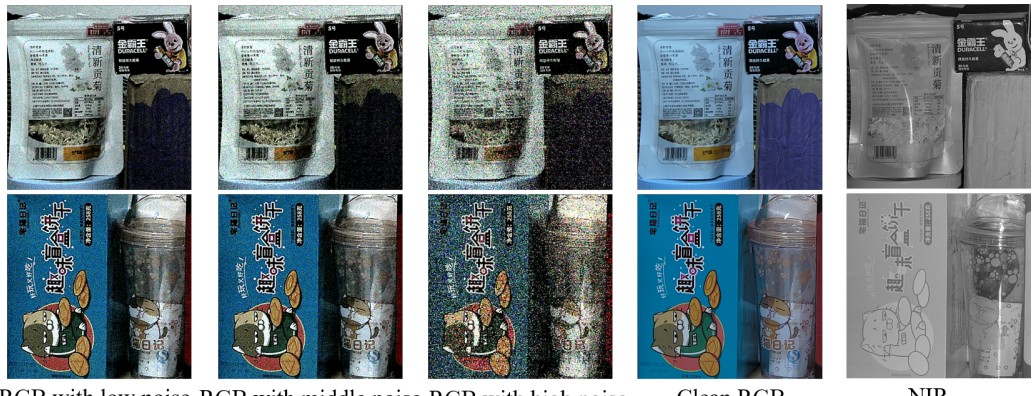

RGB with low noise RGB with middle noise RGB with high noise  Clean RGB    NIR

Figure 3: Image examples from our real-world NIR-assist image denoising (NAID) dataset.

Table 1: Comparisons of some existing datasets consisting of paired NIR and RGB images. 'Public' refers to its current public accessibility. 'Dataset Size' denotes the number of paired images.

| Dataset | Real Noise | Public | Dataset Size | Image Resolution |
|---|---|---|---|---|
| RGB-NIR Video (Cheng et al., 2023) | | | 11444 | $1280 \times 720$ |
| RGB-NIR Stereo (Wang et al., 2019a) | | ✓ | 42000 | $\sim 582 \times 492$ |
| IVRG (Brown & Süsstrunk, 2011) | | ✓ | 477 | $\sim 1024 \times 680$ |
| DVD (Jin et al., 2022) | | ✓ | 307 | $1792 \times 1008$ |
| Burst Dataset (Wang et al., 2019a) | ✓ | | 121 | $512 \times 512$ |
| VIS-NIR-MIX (Lv et al., 2020) | ✓ | | 206 | $\sim 3072 \times 2048$ |
| Dark Flash Photography (Zhi et al., 2018) | ✓ | ✓ | 5 | $\sim 1400 \times 1000$ |
| NAID (Ours) | ✓ | ✓ | 300 | $2160 \times 2048$ |

capture images of static objects. In total, the dataset comprises 100 scenes with diverse contents, and each scene has three noisy images with various noise levels. 90 scenes are randomly sampled as the training set and the remaining 10 ones are used for the testing set. In addition, we compare the NAID dataset with other existing NIR-RGB ones to demonstrate the strengths of our dataset, as shown in Table 1. Some image examples in the dataset are also provided in Fig. 3. More details about the dataset are provided in Sec. A of the appendix.

## 4 METHOD

### 4.1 PROBLEM FORMATION

NIR-assisted image denoising aims at restoring the clean RGB image $\hat{\mathbf{I}} \in \mathbb{R}^{H \times W \times 3}$ from its noisy RGB observation $\mathbf{I_R} \in \mathbb{R}^{H \times W \times 3}$ with the assistance of the NIR image $\mathbf{I_N} \in \mathbb{R}^{H \times W \times 1}$, where $H$

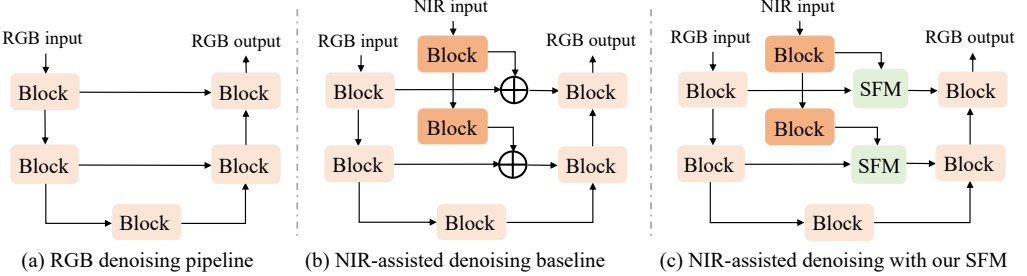

Figure 4: Comparison of different image denoising methods with multi-scale architecture. (**a**) RGB image denoising. (**b**) NIR-assisted RGB image denoising baseline. (**c**) NIR-assisted RGB image denoising with our proposed Selective Fusion Module (SFM).

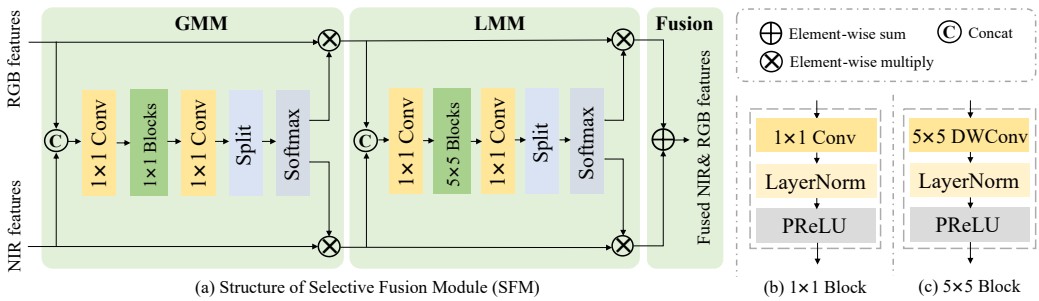

Figure 5: The structure of our proposed Selective Fusion Module (SFM), where Global Modulation Module (GMM) and Local Modulation Module (LMM) focus on color and structure discrepancy issues between the NIR images and RGB ones, respectively. Two $1 \times 1$ blocks and $5 \times 5$ blocks are used in GMM and LMM, respectively.

and $W$ denote the height and width of images, respectively. Compared to the vanilla image denoising based on the multi-scale encoder-decoder architectures as shown in Fig. 4 (a), it further plays the role of the corresponding NIR image to guide the noise removal. And that's also the core of the NIR-assisted image denoising. Assuming that the clean NIR images are perfectly consistent with the noisy RGB ones in color and structure, we can simply adapt the existing denoising architectures in Fig. 4 (a) to Fig. 4 (b). The output $\hat{\mathbf{I}}$ can be written as,

$$\hat{\mathbf{I}} = \mathcal{D}(\mathcal{E}_N(\mathbf{I_N}) + \mathcal{E}_R(\mathbf{I_R})), \tag{1}$$

where $\mathcal{D}$ denotes the decoder of the denoising network, $\mathcal{E}_N$ and $\mathcal{E}_R$ denote the feature encoders for NIR and RGB images, respectively.

However, in practical scenarios, there are color and structure inconsistencies between the NIR images and the RGB ones, as illustrated in Fig. 1. Leveraging the NIR images in a naive way like Eqn. (1) only gains limited performance improvement. Instead, we propose an Selective Fusion Module (SFM) for combining NIR-RGB information to address the issue, as shown in Fig. 4 (c). Thus, Eqn. (1) can be modified to,

$$\hat{\mathbf{I}} = \mathcal{D}(\mathcal{SFM}(\mathcal{E}_N(\mathbf{I_N}), \mathcal{E}_R(\mathbf{I_R}))). \tag{2}$$

## 4.2 SELECTIVE FUSION MODULE

SFM should select valuable information and avoid harmful one from the current NIR-RGB features for feature fusion. To achieve that, we suggest that SFM predicts and assigns pixel-wise weights for NIR-RGB features fusion. Denote the current NIR and RGB features from the corresponding encoders by $\mathbf{F_N}$ and $\mathbf{F_R}$, SFM can be written as,

$$\mathcal{SFM}(\mathbf{F_N}, \mathbf{F_R}) = \mathbf{W_N} \odot \mathbf{F_N} + \mathbf{W_R} \odot \mathbf{F_R}, \tag{3}$$

where $\odot$ is the pixel-wise multiply operation. $\mathbf{W_N}$ and $\mathbf{W_R}$ denote the weight of NIR and RGB features, respectively. In order to model the color and structure discrepancy respectively, we decouple the weight $\mathbf{W}$ (including $\mathbf{W_N}$ and $\mathbf{W_R}$) into global and local component, *i.e.*, $\mathbf{W} = \mathbf{W}^g \odot \mathbf{W}^l$,

where the former one concentrates on the differences in global information and the latter one focuses on the discrepancy in local information between NIR-RGB features. Based on that, we further present a Global Modulation Module (GMM) to estimate $\mathbf{W}^g$ and a Local Modulation Module (LMM) to estimate $\mathbf{W}^l$, as shown in Fig. 5.

**Global Modulation Module.** GMM should handle the global color and brightness difference between the NIR and RGB ones. As shown in Fig. 5 (a), it takes the current NIR features $\mathbf{F_N}$ and the RGB ones $\mathbf{F_R}$ as inputs to estimate the NIR global modulation weights $\mathbf{W_N^g}$ and the RGB ones $\mathbf{W_R^g}$. Detailly, $\mathbf{F_N}$ and $\mathbf{F_R}$ are concatenated along channel dimension followed with a $1 \times 1$ convolutional layer for channel reduction. Two $1 \times 1$ blocks are then deployed to get the deep fused feature maps, which are passed into another $1 \times 1$ convolutional layer , a channel split operation, and a softmax operation sequentially to get the estimated NIR weights $\mathbf{W_N^g}$ and RGB ones $\mathbf{W_R^g}$. Each $1 \times 1$ block is composed of a $1 \times 1$ convolutional layer, a Layer Normalization (Ba et al., 2016), and a PReLU (He et al., 2015) function, as shown in Fig. 5 (b). We modulate the NIR features $\mathbf{F_N}$ and the RGB ones $\mathbf{F_R}$ with $\mathbf{W_N^g}$ and $\mathbf{W_R^g}$, respectively, *i.e.*,

$$\mathbf{F_N^g} = \mathbf{W_N^g} \odot \mathbf{F_N}, \quad \mathbf{F_R^g} = \mathbf{W_R^g} \odot \mathbf{F_R}, \tag{4}$$

where $\mathbf{F_N^g}$ and $\mathbf{F_R^g}$ are the globally modulated NIR features and the RGB ones, respectively.

**Local Modulation Module.** The Local Modulation Module (LMM) should focus on the structure inconsistency between NIR images and RGB ones. We suggest increasing the receptive field to perceive more structure information from a range of neighboring pixels. In detail, LMM takes the globally modulated NIR features $\mathbf{F_N^g}$ and the RGB ones $\mathbf{F_R^g}$ as inputs to estimate the local NIR weights $\mathbf{W_N^l}$ and the RGB ones $\mathbf{W_R^l}$, as shown in Fig. 5 (a). Without complex network design, LMM is built upon GMM by replacing the $1 \times 1$ convolutional layer in $1 \times 1$ block to a large kernel depth-wise convolutional layer (DWConv) (Howard et al., 2017) for capturing more local information, as shown in Fig. 5 (c). Finally, $\mathbf{W_N^l}$ and $\mathbf{W_R^l}$ are employed to get the fused NIR and RGB feature $\mathbf{F_{NR}}$ as,

$$\mathbf{F_{NR}} = \mathbf{W_N^l} \odot \mathbf{F_N^g} + \mathbf{W_R^l} \odot \mathbf{F_R^g}. \tag{5}$$

$\mathbf{F_{NR}}$ is then passed to the decoder to output the denoising result.

**Discussion.** There are several advantages of our proposed SFM. First, the color and structure discrepancy issues are decoupled and addressed with GMM and LMM respectively, which achieves significant performance improvements while maintaining interpretability. Second, the compact and lightweight network design makes it only add few parameters and computation costs. Third, it is plug-and-play and can be simply integrated into existing advanced denoising networks. The related experiment results are presented in Sec. 5.

## 4.3 LOSS FUNCTION

A multi-scale loss function is adopted for updating network parameters. In detail, we employ a $3 \times 3$ convolutional layer after the decoder at scale $s$ to generate the noise-free image $\hat{\mathbf{I}}_s$. Therein, $\hat{\mathbf{I}}_1$ represents the final output with full resolution. Subsequently, we can calculate the multi-scale loss by the following formulation,

$$\mathcal{L} = \sum_{s=1}^{3} ||\hat{\mathbf{I}}_s - \mathbf{I}_{\downarrow 2^{s-1}}||_2, \tag{6}$$

where $\mathbf{I}_{\downarrow 2^{s-1}}$ denote the ground truth after $\times 2^{s-1}$ down-sampling.

## 5 EXPERIMENTS

### 5.1 EXPERIMENTAL SETTINGS

**Datasets.** Experiments are conducted on the synthetic and our real-world NAID datasets. The details of the real-world NAID dataset can be seen in Sec. 3. In addition, We use the DVD (Jin et al., 2022) dataset to generate synthetic noisy images. It comprises 307 pairs of clean RGB images (and corresponding RAW images) and NIR images. 267 pairs are used for training and 40 pairs are for testing. The way to simulate noisy data follows DVD (Jin et al., 2022). We first scale the mean value of the clean RAW images, getting synthetic low-light clean RAW images. Then we add Gaussian

noise with the variance $\sigma$ and Poisson noise with a noise level $\sigma$ to the generated low-light images. Finally, the synthetic low-light noisy RAW images are converted to RGB ones for training models. We conduct experiments with $\sigma = 4$ and $\sigma = 8$ (the larger the $\sigma$, the heavier the noise).

**Implementation Details.** We build our NIR-assisted denoising models by incorporating the proposed SFM into a CNN-based advanced denoising network (*i.e.*, NAFNet (Chen et al., 2022)) and two Transformer-based ones (*i.e.*, Uformer (Wang et al., 2022) and Restormer (Zamir et al., 2022)), which are dubbed **NIR-NAFNet**, **NIR-Uformer**, and **NIR-Restormer**, respectively. All models are trained by the Adam (Kingma & Ba, 2014) optimizer with $\beta_1 = 0.9$ and $\beta_2 = 0.999$ for 120k iterations. The batch size is set to 32 and the patch size is set to $128 \times 128$. For synthetic image denoising, the cosine annealing strategy (Loshchilov & Hutter, 2017) is employed to steadily decrease the learning rate from $2 \times 10^{-4}$ to $1 \times 10^{-6}$. For real-world image denoising, the initial learning rate is set to $3 \times 10^{-4}$ and halved every 20k iterations. All experiments are conducted with PyTorch (Paszke et al., 2019) on an Nvidia GeForce RTX A6000 GPU.

## 5.2 COMPARISON WITH STATE-OF-THE-ART METHODS

Experiments are conducted by comparing our NIR-NAFNet, NIR-Uformer, and NIR-Restormer with 8 models, including 3 single image denoising methods (*i.e.* NAFNet (Chen et al., 2022), Uformer (Wang et al., 2022), and Restormer (Zamir et al., 2022)) and 5 NIR-assisted denoising methods (*i.e.* FGDNet (Sheng et al., 2022), SANet (Sheng et al., 2023), CUNet (Deng & Dragotti, 2020), MNNet (Xu et al., 2022a), and DVN (Jin et al., 2022)). To quantitatively evaluate the performance, we calculate three metrics on the RGB channels, *i.e.* Peak Signal to Noise Ratio (PSNR) (Huynh-Thu & Ghanbari, 2008), Structural Similarity (SSIM) (Wang et al., 2004) and Learned Perceptual Image Patch Similarity (LPIPS) (Zhang et al., 2018b). We also evaluate the inference cost of different models. The #FLOPs when processing a $128 \times 128$ patch and the inference time when feeding a $1792 \times 1008$ image are reported.

**Results on synthetic DVD dataset.** The quantitative results on the synthetic DVD dataset are shown in Table 2. It can be observed that our method significantly improves performance against single-image denoising methods, thereby demonstrating the effectiveness of NIR images. In comparison with existing NIR-assisted denoising ones, our methods also outperform by a large margin, as the proposed SFM overcomes the discrepancy issues between the NIR-RGB images while coupling with the advanced denoising backbone successfully. In particular, our NIR-NAFNet makes a better trade-off between performance and efficiency than other methods. Besides, The qualitative results in Fig. 6 show that our methods restore more realistic textures and fewer artifacts than others.

**Results on real-world NAID dataset.** Real-world data has much more complex degradation than synthetic ones. The quantitative results in Table 3 show that our methods still keep high performance in the real world. Taking NIR-Restormer as an example, our proposed NIR-Restormer achieves 0.33dB, 0.54dB, and 0.94dB PSNR gains than Restormer (Zamir et al., 2022) in dealing with low-level, middle-level and high-level noise respectively. The higher the level of noise, the greater the improvement achieved by our method, which further indicates the advantage of the utilization of NIR information for low-light noise removal. The qualitative results in Fig. 7 demonstrate that our models still recover fine-scale details in the real world, while other NIR-assisted denoising methods may produce artifacts. More visual comparisons can be seen in Sec. D of the appendix.

## 6 ABLATION STUDY

We conduct ablation studies on our real-world NAID dataset when taking NIR-NAFNet as an example. They include the effect of GMM and LMM, the effect of kernel size of DWConv in LMM, and the effect of number of SFM. The metrics are reported by averaging these on three noise levels.

### 6.1 EFFECT OF GMM AND LMM IN SFM.

As shown in Table 4, the incorporation of global and local feature modulation yields 0.13dB PSNR improvements each, which can be attributed to their effective handling of the inconsistencies between NIR-RGB images in color and structure, respectively. And with both global and local feature modulation, it achieves results with 0.23dB PSNR gain. Additionally, GMM and LMM are both

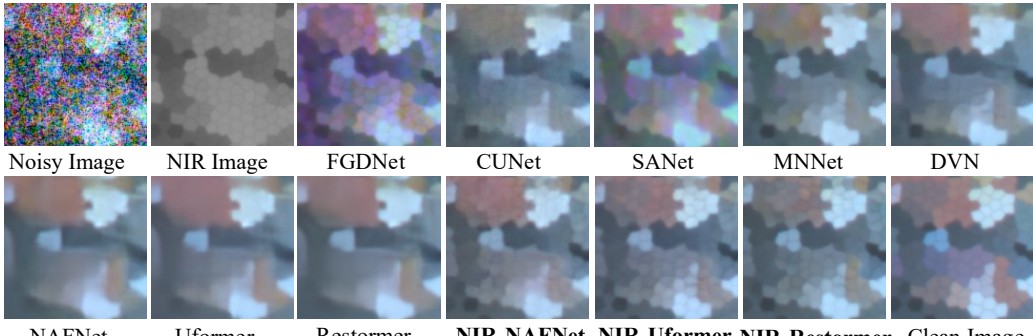

| Noisy Image | NIR Image | FGDNet | CUNet | SANet | MNNet | DVN |
| NAFNet | Uformer | Restormer | **NIR-NAFNet** | **NIR-Uformer** | **NIR-Restormer** | Clean Image |

Figure 6: Qualitative comparison on the synthetic DVD dataset. **Bold** marks our methods.

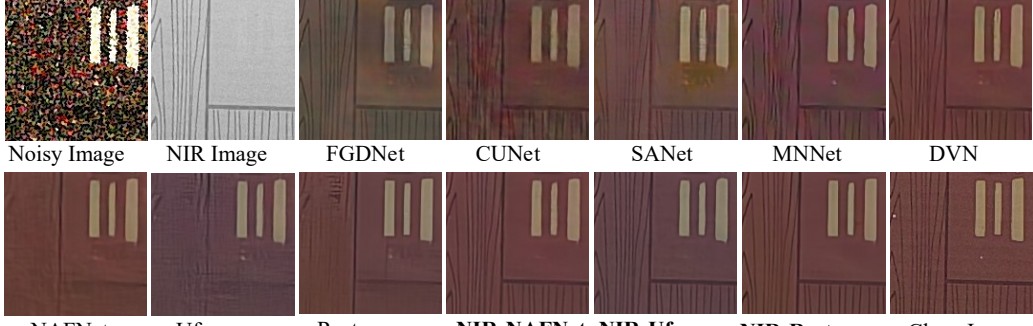

| Noisy Image | NIR Image | FGDNet | CUNet | SANet | MNNet | DVN |
| NAFNet | Uformer | Restormer | **NIR-NAFNet** | **NIR-Uformer** | **NIR-Restormer** | Clean Image |

Figure 7: Qualitative comparison on our real-world NAID dataset. **Bold** marks our methods.

Table 2: Quantitative comparison on the synthetic DVD dataset. **Bold** marks our results.

| | Methods | $\sigma = 4$ | $\sigma = 8$ | #FLOPs (G) | Time (ms) |
|---|---|---|---|---|---|
| | | PSNR↑/ SSIM↑/ LPIPS↓ | PSNR↑ / SSIM↑ / LPIPS↓ | | |
| Single-Image Denoising | Uformer (CVPR'22) | 29.58 / 0.8967 / 0.271 | 27.36 / 0.8632 / 0.352 | 19.16 | 1748 |
| | Restormer (CVPR'22) | 29.67 / 0.9038 / 0.262 | 27.41 / 0.8741 / 0.343 | 70.59 | 2048 |
| | NAFNet (ECCV'22) | 29.49 / 0.8959 / 0.263 | 27.29 / 0.8638 / 0.336 | 8.10 | 312 |
| NIR-Assisted Denoising | FGDNet (TMM'22) | 23.91 / 0.8371 / 0.439 | 22.02 / 0.7374 / 0.436 | 38.67 | 479 |
| | SANet (CVPR'23) | 27.68 / 0.8648 / 0.343 | 25.28 / 0.8304 / 0.413 | 161.06 | 2763 |
| | CUNet(TPAMI'20) | 28.01 / 0.8558 / 0.332 | 26.07 / 0.8182 / 0.412 | 14.48 | 542 |
| | MNNet (IF'22) | 28.48 / 0.8994 / 0.274 | 26.33 / 0.8697 / 0.353 | 23.68 | 1360 |
| | DVN (AAAI'22) | 29.69 / 0.9062 / 0.236 | 27.43 / 0.8799 / 0.292 | 104.50 | 761 |
| | **NIR-Uformer** | **30.10 / 0.9188 / 0.192** | **28.03 / 0.9008 / 0.238** | **24.85** | **2500** |
| | **NIR-Restormer** | **30.22 / 0.9209 / 0.193** | **28.11 / 0.8701 / 0.260** | **89.17** | **2747** |
| | **NIR-NAFNet** | **30.08 / 0.9005 / 0.208** | **27.86 / 0.8664 / 0.273** | **13.17** | **462** |

Table 3: Quantitative comparison on our real-world NAID dataset. **Bold** marks our results.

| | Methods | Low-Level Noise | Middle-Level Noise | High-Level Noise |
|---|---|---|---|---|
| | | PSNR↑ / SSIM↑ / LPIPS↓ | PSNR↑ / SSIM↑ / LPIPS↓ | PSNR↑ / SSIM↑ / LPIPS↓ |
| Single-Image Denoising | Uformer (CVPR'22) | 25.56 / 0.7736 / 0.304 | 24.52 / 0.7418 / 0.347 | 23.31 / 0.7091 / 0.389 |
| | Restormer (CVPR'22) | 25.89 / 0.7842 / 0.294 | 24.98 / 0.7572 / 0.333 | 23.82 / 0.7297 / 0.387 |
| | NAFNet (ECCV'22) | 25.71 / 0.7780 / 0.294 | 24.76 / 0.7482 / 0.335 | 23.71 / 0.7186 / 0.378 |
| NIR-Assisted Denoising | FGDNet (TMM'22) | 24.25 / 0.7676 / 0.368 | 22.89 / 0.7367 / 0.430 | 21.86 / 0.7080 / 0.509 |
| | CUNet (TPAMI'20) | 24.05 / 0.7314 / 0.313 | 23.29 / 0.7031 / 0.380 | 22.41 / 0.6398 / 0.449 |
| | SANet (CVPR'23) | 24.93 / 0.7679 / 0.359 | 23.74 / 0.7335 / 0.416 | 22.69 / 0.7028 / 0.476 |
| | MNNet (IF'22) | 25.68 / 0.7797 / 0.313 | 24.64 / 0.7512 / 0.364 | 23.36 / 0.7194 / 0.419 |
| | DVN (AAAI'22) | 25.96 / 0.7853/ 0.298 | 24.93 / 0.7578 / 0.332 | 23.95 / 0.7360 / 0.382 |
| | **NIR-Uformer** | **25.91 / 0.7919 / 0.276** | **25.14 / 0.7714 / 0.299** | **24.28 / 0.7534 / 0.321** |
| | **NIR-Restormer** | **26.22 / 0.7963 / 0.265** | **25.51 / 0.7767 / 0.293** | **24.76 / 0.7626 / 0.315** |
| | **NIR-NAFNet** | **26.06 / 0.7905 / 0.274** | **25.26 / 0.7676 / 0.303** | **24.48 / 0.7503 / 0.321** |

Table 4: Quantitative comparison with different modulation modules in SFM.

| GMM | LMM | PSNR↑ / SSIM↑ / LPIPS↓ |
|-----|-----|------------------------|
| × | × | 25.03 / 0.7647 / 0.304 |
| ✓ | × | 25.16 / 0.7675 / 0.302 |
| × | ✓ | 25.16 / 0.7653 / 0.302 |
| ✓ | ✓ | 25.26 / 0.7695 / 0.299 |

Table 5: Quantitative comparison of different arrangements of GMM and LMM.

| Arrangement | PSNR↑ / SSIM↑ / LPIPS↓ |
|-------------|------------------------|
| GMM + GMM | 25.17 / 0.7652 / 0.302 |
| LMM + LMM | 25.18 / 0.7650 / 0.301 |
| LMM + GMM | 25.19 / 0.7669 / 0.301 |
| GMM + LMM | 25.26 / 0.7695 / 0.299 |

Table 6: Quantitative comparison of different numbers of SFM at each scale.

| #SFM | PSNR↑ / SSIM↑ / LPIPS↓ |
|------|------------------------|
| 1 | 25.26 / 0.7695 / 0.299 |
| 3 | 25.28 / 0.7699 / 0.299 |
| 5 | 25.29 / 0.7669 / 0.301 |

Table 7: Quantitative comparison of different kernel sizes of DWConv in LMM.

| Kernel Size | PSNR↑ / SSIM↑ / LPIPS↓ |
|-------------|------------------------|
| $3 \times 3$ | 25.22 / 0.7685 / 0.298 |
| $5 \times 5$ | 25.26 / 0.7695 / 0.299 |
| $7 \times 7$ | 25.28 / 0.7717 / 0.301 |

lightweight modules that do not increase the number of parameters and inference time too much. The number of parameters of GMM and LMM only account for $1.5\%$ and $1.1\%$ of those of NIR-NAFNet, respectively. Applying an SFM on NIR-NAFNet only results in a time increase of 1 ms. More results compared to the NIR-assisted image denoising baseline in Fig. 4 (b) with different denoising backbones can be seen in Sec. B of the appendix.

To further demonstrate the effectiveness of decoupling the inconsistencies into color and structure components, we conduct experiments with different arrangements of GMM and LMM. The results are shown in Table 5. 'GMM + GMM' and 'LMM + LMM' mean that modulate features with 2 GMMs and 2 LMMs respectively, but results in limited performance gain. This shows that our performance improvement is not due to a simple increase in parameter numbers. 'LMM + GMM' denotes that modulates features first locally and then globally, also leading to limited improvement. It may be because the significant difference in global content leads to inaccurate local feature modulation. Therefore, we deploy a GMM to handle color discrepancy first followed by a LMM dealing with structure discrepancy, dubbed 'GMM + LMM', achieving better results.

## 6.2 EFFECT OF KERNEL SIZE OF DWCONV IN LMM.

To illustrate the effect of the size of receptive fields in LMM, we conduct experiments employing varying kernel sizes of DWConv as shown in Table 7. Generally, a larger kernel size leads to greater performance improvement, which proves that a large reception field helps the local modulation of features. But it is improved marginally when the kernel size is larger than $5 \times 5$. For the sake of simplicity and efficiency, we set the kernel size of DWConv to $5 \times 5$ as default.

## 6.3 EFFECT OF NUMBER OF SFM.

Here we investigate the effect of incorporating different numbers of SFM into the NAFNet (Chen et al., 2022) at each scale. The results are shown in Table 6. It can be observed that the performance generally increases marginally as the number of SFMs grows. Also for the sake of simplicity and efficiency, we only set the number of SFM to 1 at each scale of the networks.

## 7 CONCLUSION

Near-infrared (NIR) images can help restore fine-scale details while removing noise from noisy RGB images, especially in low-light environments. The content inconsistency between NIR-RGB images and the scarcity of real-world paired datasets limit its effective application in real scenarios. In this work, we propose a plug-and-play Selective Fusion Module (SFM) and a real-world paired NIR-Assisted Image Denoising (NAID) dataset to address these issues. Specifically, SFM sequentially performs global and local modulations on NIR-RGB features before their information fusion. The NAID dataset is collected with various noise levels under diverse scenes. Experiments on both synthetic and real-world datasets show our method achieves better results than state-of-the-art ones.

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

APPENDIX

The content of the appendix involves:

- More details about the NAID dataset in Sec. A
- More comparisons with NIR-assisted image denoising baseline in Sec. B
- Effect of GMM and LMM in SFM in Sec. C
- More qualitative comparisons in Sec. D

## A  MORE DETAILS ABOUT THE NAID DATASET

All images in the NAID dataset are captured with the Huawei X2381-VG camera. It is a common surveillance camera equipped with a built-in NIR illuminator specifically designed for capturing NIR images. When capturing clean RGB images, the camera's ISO is set to 600. When capturing RGB images with low, middle, and high noise levels, we set ISO to about 4000, 12000, and 32000 respectively. It is worth noting that we adjust the exposure time of each noise level to keep the brightness of noisy images relatively constant. Besides, following DVD (Jin et al., 2022), we crop all images from $2048 \times 3840$ resolutions into $2048 \times 2160$ to mitigate the vignetting effect.

## B  MORE COMPARISONS WITH NIR-ASSISTED IMAGE DENOISING BASELINE

We incorporate our SFM into two Transformer-based denoising networks (*i.e.*, Uformer (Wang et al., 2022) and Restormer (Zamir et al., 2022)) and a CNN-based one (*i.e.*, NAFNet (Chen et al., 2022)), dubbed NIR-Uformer, NIR-Restormer, and NIR-NAFNet respectively. Uformer (Wang et al., 2022) and Restormer (Zamir et al., 2022) are different representations of Transformer-based models, where the former calculates self-attention in spatial dimension while the latter in the channel one. As shown in Fig. 4 (b), we sum the NIR features and RGB ones as our baseline, *i.e.*, 'Uformer-Baseline', 'Restormer-Baseline', and 'NAFNet-Baseline', respectively. From Table A and B, it is observed that the performance of different denoising networks can be further improved by integrating our proposed SFM on both synthetic DVD and real-world NVID datasets. It further validates the effectiveness of our proposed SFM and its plug-and-play nature.

## C  EFFECT OF GMM AND LMM IN SFM

We provide the qualitative results of GMM and LMM in SFM taking NIR-NAFNet as an example. As shown in Fig. A, the incorporation of our GMM and LMM in SFM both helps fine-scale texture recovery. Please zoom in for more clear observation.

## D  MORE QUALITATIVE COMPARISONS

We provide more qualitative results by comparing our NIR-NAFNet, NIR-Uformer, and NIR-Restormer with 8 models, including 3 single image denoising methods (*i.e.*, NAFNet (Chen et al., 2022), Uformer (Wang et al., 2022), and Restormer (Zamir et al., 2022)) and 5 NIR-assisted denoising ones (*i.e.*, FGDNet (Sheng et al., 2022), SANet (Sheng et al., 2023), CUNet (Deng & Dragotti, 2020), MNNet (Xu et al., 2022a), and DVN (Jin et al., 2022)). As shown in Fig. B and Fig. C, our methods restore more fine-scale and photo-realistic textures than others, both on synthetic and real-world images.

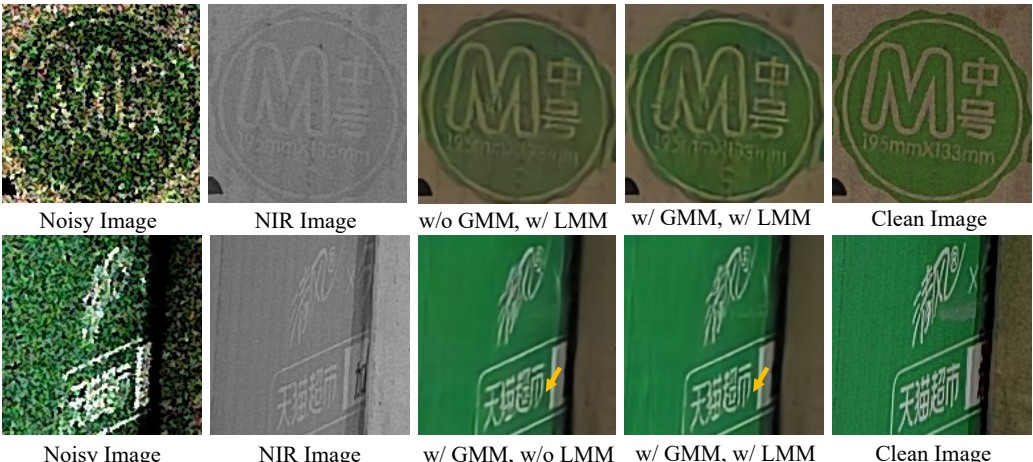

| Noisy Image | NIR Image | w/o GMM, w/ LMM | w/ GMM, w/ LMM | Clean Image |
| Noisy Image | NIR Image | w/ GMM, w/o LMM | w/ GMM, w/ LMM | Clean Image |

Figure A: The qualitative comparisons of incorporating GMM and LMM to modulate features. We mark some areas with yellow arrows for easier observation.

Table A: Comparisons of our proposed method with the NIR-assisted image denoising baseline (see Fig. 4 (b)) on different networks on synthetic DVD dataset.

| | Methods | $\sigma = 4$ | $\sigma = 8$ | #FLOPs | Time |
|---|---|---|---|---|---|
| | | PSNR↑/ SSIM↑/ LPIPS↓ | PSNR↑/ SSIM↑/ LPIPS↓ | (G) | (ms) |
| Spatial-Wised | Uformer-Baseline | 29.74 / 0.9156 / 0.215 | 27.81 / 0.8917 / 0.264 | 23.39 | 2282 |
| Transformer-Based | NIR-Uformer | 30.10 / 0.9188 / 0.192 | 28.03 / 0.9008 . 0.238 | 24.85 | 2500 |
| Channel-Wised | Restormer-Baseline | 29.97 / 0.8897 / 0.199 | 27.88 / 0.8680 / 0.245 | 86.75 | 2578 |
| Transformer-Based | NIR-Restormer | 30.22 / 0.9209 / 0.193 | 28.11 / 0.8701 / 0.260 | 89.17 | 2747 |
| CNN-Based | NAFNet-Baseline | 29.70 / 0.8894 / 0.220 | 27.62 / 0.8621 / 0.276 | 11.72 | 461 |
| | NIR-NAFNet | 30.08 / 0.9005 / 0.208 | 27.86 / 0.8664 / 0.273 | 13.17 | 462 |

Table B: Comparisons of our proposed method with the NIR-assisted image denoising baseline (see Fig. 4 (b)) on different networks on our real-world NVID dataset.

| | Methods | Low-Level Noise | Middle-Level Noise | High-Level Noise |
|---|---|---|---|---|
| | | PSNR↑/ SSIM↑/ LPIPS↓ | PSNR↑/ SSIM↑/ LPIPS↓ | PSNR↑/ SSIM↑/ LPIPS↓ |
| Spatial-Wised | Uformer-Baseline | 25.80 / 0.7904 / 0.269 | 25.03 / 0.7687 / 0.294 | 24.02 / 0.7488 / 0.320 |
| Transformer-Based | NIR-Uformer | 25.91 / 0.7917 / 0.276 | 25.14 / 0.7714 / 0.299 | 24.28 / 0.7534 / 0.321 |
| Channel-Wised | Restormer-Baseline | 26.12 / 0.7948 / 0.265 | 25.38 / 0.7733 / 0.293 | 24.61 / 0.7587 / 0.316 |
| Transformer-Based | NIR-Restormer | 26.22 / 0.7963 / 0.265 | 25.51 / 0.7767 / 0.293 | 24.76 / 0.7626 / 0.315 |
| CNN-Based | NAFNet-Baseline | 25.92 / 0.7883 / 0.277 | 25.00 / 0.7628 / 0.304 | 24.16 / 0.7429 / 0.329 |
| | NIR-NAFNet | 26.06 / 0.7905 / 0.274 | 25.26 / 0.7676 / 0.303 | 24.48 / 0.7503 / 0.321 |

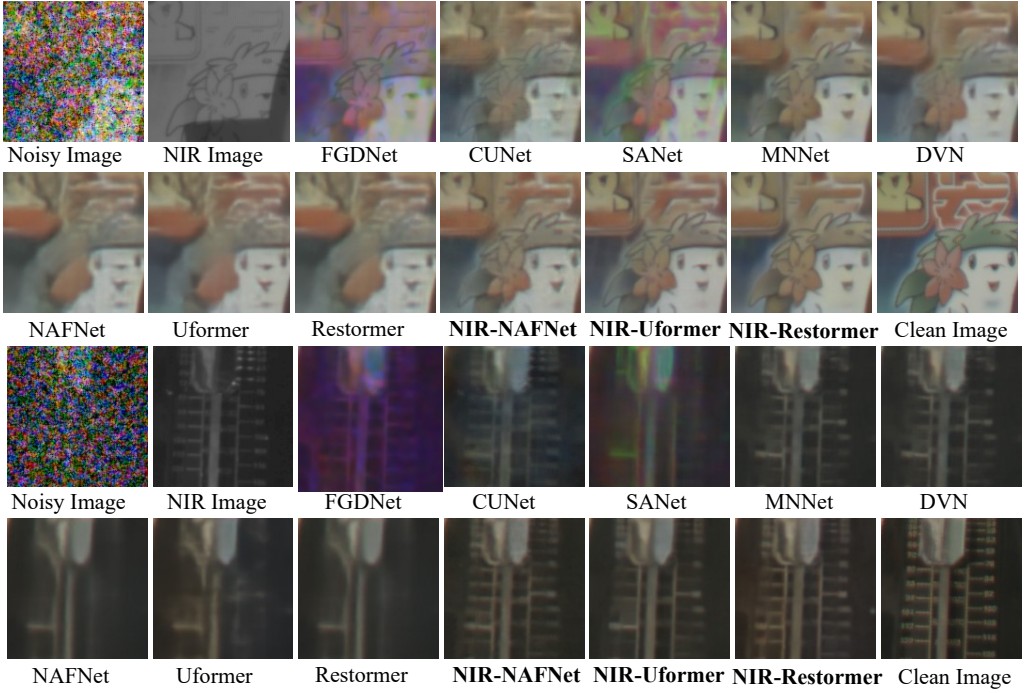

Figure B: More qualitative comparisons on synthetic DVD dataset. **Bold** marks our methods.

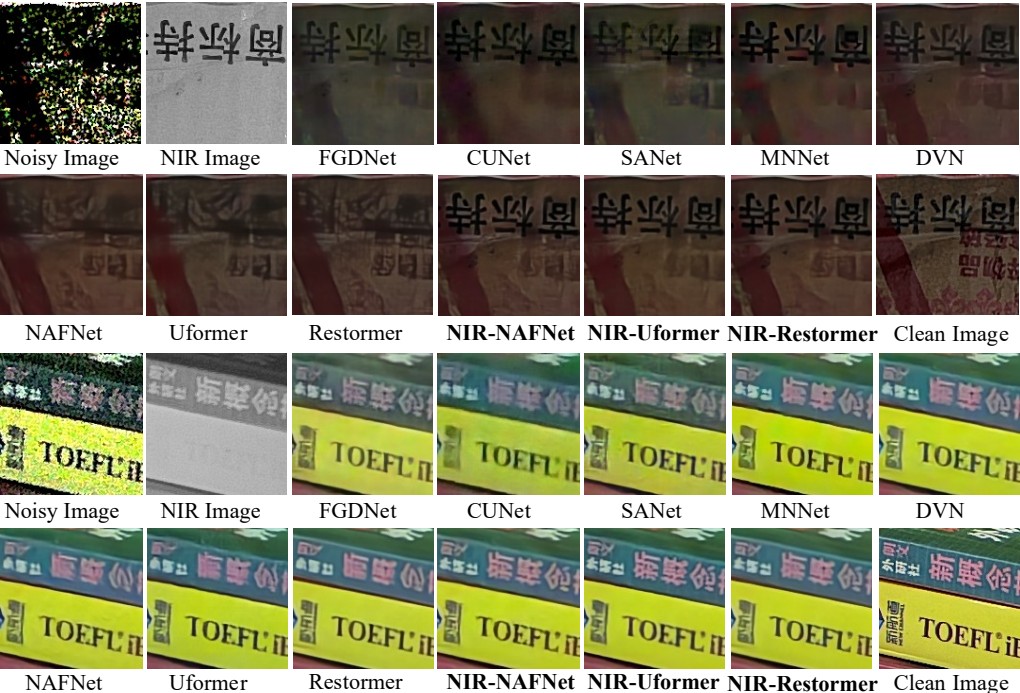

Figure C: More qualitative comparisons on real-world NVID dataset. **Bold** marks our methods.

