# OpenReview forum: "NIR-Assisted Image Denoising: A Selective Fusion Approach and A Real-World Benchmark Dataset"
_ICLR.cc/2024/Conference — ICLR 2024 Conference Withdrawn Submission_

### Official Review · Reviewer_KiWB · 2023-10-30

**Soundness:** 2 fair
**Presentation:** 2 fair
**Contribution:** 2 fair
**Rating:** 5
**Confidence:** 4

**Summary:**

This paper proposes an NIR-assist RGB image denoising method, which mainly includes two parts: First, an effective SFM module is proposed to help the selective fusion of NIR and RGB images and solve the content inconsistency between NIR-RGB images. And the module is plug-and-play. Secondly, a real-world NIR-Assisted Image Denoising (NAID) dataset is proposed.

**Strengths:**

The authors conduct experiments to verify the proposed method for denoising images on both synthetic and proposed real-world datasets. Ablation studies are conducted to validate the effectiveness of the two contributions. The paper looks technically sound and describes the algorithm clearly.

**Weaknesses:**

I am concerned that the overall contributions are trivial. Especially, the proposed SFM module should be provided with more design reasons.

Weaknesses:

1. For GMM and LMM, NIR and RGB are concatenated along channel dimensions to input the same subsequent modules, and then split to obtain the two estimated NIR weights and RGB weights to apply to the corresponding branches, respectively. It seems to me that two estimated weights are the same, how to achieve selective fusion? Or are manually setting parameters involved? It would be better to provide some justification or motivation for the design choices.
2. Table 7 is not mentioned in the paper.

**Questions:**

See Weaknesses

---

> ### Author Response · Authors · 2023-11-22
> **Response to Reviewer KiWB (1/2)**
>
> We thank the reviewer for the valuable comments and suggestions. We appreciate the reviewer's questions and hope our responses can address the concerns.
>
> `1.` **Overall contributions.**
>
> Previous related works perform NIR-assist image denoising on synthetic noisy images. The most important contribution of this work is to introduce this into the real world. We will revise the paper to make the contribution clear.
>
> Moreover, other main contributions can be summarized as follows:
>
> (1) Towards the content inconsistency issue, we propose a simple yet efficient selective fusion module (SFM) that can be seamlessly integrated into existing denoising networks with few increasing computation costs.
>
> (2) Towards the real-world dataset scarcity issue, we construct a real-world NIR-assisted image denoising (NAID) dataset that covers diverse scenarios and various noise levels, which is expected to serve as a benchmark for future works.
>
> (3) Extensive experiments demonstrate that our method achieves better results than state-of-the-art ones.
>
> In addition, we promise the dataset, codes, and pre-trained models will be publicly available, which may be advantageous for future studies.
>
> `2.` **Motivation of the SFM module.**
>
> Color and structure inconsistency issues between NIR-RGB images are the main challenges in NIR-assisted image denoising. First, the NIR image is captured under additional NIR light and is monochromatic, which leads to color discrepancy between NIR-RGB images. Second, the NIR image may 'more-see' or 'less-see' the objects than the visible light ones, primarily due to inherent differences in the optical properties within each spectral domain, which leads to structure discrepancy between NIR-RGB images.
>
> On the one hand, we hope to design a module to address two inconsistency issues in this work. Thus, we present a global modulation module (GMM) and a local modulation module (LMM) to handle the color and structure inconsistencies, respectively. The combination of the two modules forms SFM. On the other hand, we hope the module can be simple, efficient, and plug-and-play. Thus, we intentionally avoid some complex operations (e.g., self-attention and cross-attention) during design in order to maintain efficiency as much as possible.
>
> In addition, in the second response of '**Response to Reviewer YVkk**', we further elaborate on the advantages and characteristics of the proposed SFM.
>
> We will revise the relevant descriptions to make SFM clearer in the revision.

---

> ### Author Response · Authors · 2023-11-22
> **Response to Reviewer KiWB (2/2)**
>
> `3.` **How to achieve selective fusion?**
>
> We apologize for the misunderstanding caused by the unclear description of the proposed SFM. Here we provide a more detailed and clear description.
>
> SFM consists of a GMM, an LMM, and a fusion operation, where GMM mainly handles the color discrepancy and LMM handles the structure one. It is worth noting that SFM automatically predicts the weights for RGB and NIR features, without manually setting parameters. And the selective fusion is mainly inspired by ‘Selective Kernel Networks’[a].
>
> Specifically, for GMM, it first concatenates NIR features $\mathbf{F_{N}} \in \mathbb{R}^{C \times H \times W}$ and RGB ones $\mathbf{F_{R}} \in \mathbb{R}^{C \times H \times W}$ as inputs, generating initial weights $\mathbf{W_{init}^{g}} \in \mathbb{R}^{2C \times H \times W}$ by a $1\times1$ convolutional layer, two $1\times1$ blocks for feature enhancement and another $1\times1$ convolutional layer. It can be written as,
> $$\mathbf{W_{init}^{g}} = \mathrm{Conv_{1\times1}}(\mathrm{Blocks_{1\times1}}(\mathrm{Conv_{1\times1}}([\mathbf{F_{N}},\mathbf{F_{R}}]))),$$
>
> where $\mathrm{Conv_{1\times1}}$ denotes the $1\times1$ convolutional layer and $\mathrm{Blocks_{1\times1}}$ denotes two $1\times1$ blocks. Next, $\mathbf{W_{init}^{g}}$ is split along channel dimension to get NIR weight maps $\mathbf{\tilde{W} _{N}^{g}} \in \mathbb{R}^{C \times H \times W}$ and RGB ones $\mathbf{\tilde{W} _{R}^{g}} \in \mathbb{R}^{C \times H \times W}$. Then, a
> channel-wise softmax operation is applied to  $\mathbf{\tilde{W} _{N}^{g}}$ and  $\mathbf{\tilde{W} _{R}^{g}}$, getting the final NIR modulation weights $\mathbf{W}^g _\mathbf{N} \in \mathbb{R}^{C \times H \times W}$ and RGB ones $\mathbf{W}^g _\mathbf{R} \in \mathbb{R}^{C \times H \times W}$, which can be written as,
> $$ [\mathbf{W}^g _\mathbf{N}] _{i} = \frac{\exp([\mathbf{\tilde{W}}^g _\mathbf{N}] _{i})}{\exp([\mathbf{\tilde{W}}^g _\mathbf{N}] _{i}) + \exp([\mathbf{\tilde{W}}^g _\mathbf{R}] _{i})},  \qquad[\mathbf{W}^g _\mathbf{R}] _{i} = \frac{\exp([\mathbf{\tilde{W}}^g _\mathbf{R}] _{i})}{\exp([\mathbf{\tilde{W}}^g _\mathbf{N}] _{i}) + \exp([\mathbf{\tilde{W}}^g _\mathbf{R}] _{i})}.$$
>
> where $[\ \cdot\ ] _{i}$ denotes the selection of the $i$-th channel. Finally, we modulate the NIR features $\mathbf{F} _\mathbf{N}$ and the RGB ones $\mathbf{F} _\mathbf{R}$ with $\mathbf{W}^g _\mathbf{N}$ and $\mathbf{W}^g _\mathbf{R}$, respectively, i.e.,
> $$
> \mathbf{F}^g _\mathbf{N} = \mathbf{W}^g _\mathbf{N} \odot \mathbf{F _{N}},
> \quad
> \mathbf{F}^g _\mathbf{R} = \mathbf{W}^g _\mathbf{R} \odot \mathbf{F _{R}},
> $$
>
> where $\mathbf{F}^g _\mathbf{N} \in \mathbb{R}^{C \times H \times W}$ and $\mathbf{F}^g _\mathbf{R} \in \mathbb{R}^{C \times H \times W}$ are the globally modulated NIR features and the RGB ones, respectively.
>
> In order to keep simplicity and capture more local information, LMM is built upon GMM by replacing the $1\times1$ convolutional layer in $1\times1$ blocks with a $5\times5$ depth-wise convolutional layer. As its processing is similar to GMM, we do not elaborate on it cumbersomely.
>
> We will revise the relevant descriptions to make selective fusion clearer in the revision.
>
> `4.` **Table 7 is not mentioned in the paper.**
>
> Thanks for pointing it out, we will rectify it in the revision by illustrating Table 7 in Sec. 6.2.
>
> Reference
>
> [a] Li, Xiang, et al. "Selective kernel networks." CVPR. 2019.

---

### Official Review · Reviewer_D7EG · 2023-10-31

**Soundness:** 3 good
**Presentation:** 3 good
**Contribution:** 3 good
**Rating:** 5
**Confidence:** 3

**Summary:**

This paper proposes an efficient Selective Fusion Module (SFM) for NIR-Assisted Image Denoising (NAID) and presents a real-world dataset covering diverse scenarios and noise levels. The proposed method achieves better results than state-of-the-art techniques, and the dataset serves as a benchmark for future research. The SFM decouples color and structure discrepancy issues and addresses them with Gaussian Mixture Model (GMM) and Laplacian Mixture Model (LMM), respectively. The compact and lightweight network design adds few parameters and computation costs and can be integrated into existing advanced denoising networks.

**Strengths:**

1. The proposed SFM achieves significant performance improvements while maintaining interpretability.
2. The NAID dataset covers diverse scenarios and noise levels, making it a valuable benchmark for future research.
3. The compact and lightweight network design adds few parameters and computation costs and can be integrated into existing advanced denoising networks.

**Weaknesses:**

1. Relative to other NIR-assist denoising approaches, the innovation quotient of the proposed method appears constrained, potentially limiting its adaptability across diverse imaging conditions and noise levels.
2. Concerns arise regarding the efficiency of the methodology, particularly when juxtaposed with alternative techniques, as evidenced by its prolonged training durations.
3. The introduced SFM module lacks explicit clarity on addressing pivotal challenges inherent to NIR-assist operations.

**Questions:**

1.How does the proposed SFM compare to other denoising techniques in terms of computational efficiency?
2.Can the proposed method be applied to other types of images, such as medical images or satellite images?
3.How does the NAID dataset compare to other benchmark datasets in terms of diversity and size?

---

> ### Author Response · Authors · 2023-11-22
> **Response to Reviewer D7EG (1/2)**
>
> We thank the reviewer for the valuable comments and suggestions. We appreciate the reviewer's questions and hope our responses can address the concerns.
>
> `1.` **Overall contributions and innovation quotient.**
>
> Previous related works perform NIR-assist image denoising on synthetic noisy images. We kindly remind the reviewer that the most important contribution of this work is to introduce this into the real world. We will revise the paper to make the contribution clear.
>
> Moreover, other main contributions can be summarized as follows:
>
> (1) Towards the content inconsistency issue, we propose a simple yet efficient selective fusion module (SFM) that can be seamlessly integrated into existing denoising networks with few increasing computation costs.
>
> (2) Towards the real-world dataset scarcity issue, we construct a real-world NIR-assisted image denoising (NAID) dataset that covers diverse scenarios and various noise levels, which is expected to serve as a benchmark for future works.
>
> (3) Extensive experiments demonstrate that our method achieves better results than state-of-the-art ones.
>
> In addition, we promise the dataset, codes, and pre-trained models will be publicly available, which may be advantageous for future studies.
>
> `2.` **The introduced SFM module lacks explicit clarity on addressing pivotal challenges inherent to NIR-assist operations.**
>
> Here we elaborate on the motivations of SFM.
>
> Color and structure inconsistency issues between NIR-RGB images are the main challenges in NIR-assisted image denoising. First, the NIR image is captured under additional NIR light and is monochromatic, which leads to color discrepancy between NIR-RGB images. Second, the NIR image may 'more-see' or 'less-see' the objects than the visible light ones, primarily due to inherent differences in the optical properties within each spectral domain, which leads to structure discrepancy between NIR-RGB images.
>
> On the one hand, we hope to design a module to address two inconsistency issues in this work. Thus, we present a global modulation module (GMM) and a local modulation module (LMM) to handle the color and structure inconsistencies, respectively. The combination of the two modules forms SFM. Moreover, the visualization results in Figure A of the appendix show that GMM indeed enables better color and LMM indeed enables better structure.
>
> On the other hand, we hope the module can be simple, efficient, plug-and-play. Thus, we intentionally avoid some complex operations (e.g., self-attention and cross-attention) during design in order to maintain efficiency as much as possible.
>
> In addition, in the second response of '**Response to Reviewer YVkk**', we further elaborate on the advantages and characteristics of the proposed SFM.
>
> We will revise the relevant descriptions to make SFM clearer in the revision.
>
> `3.` **Training and inference efficiency.**
>
> We compare the proposed method with the most competitive one DVN [a] as well as the naive feature fusion manner (i.e., feature summation). It is worth noting that we retrain all compared methods with the same configurations, thus the comparisons are fair. As shown in the following table, our method achieves higher performance with lower training time than the compared ones, which illustrates the training efficiency of the proposed method.
> | Methods | Training Time | Inference TIme | PSNR / SSIM / LPIPS |
> |:---:|:---:|:---:|:---:|
> | DVN | 44h 50min | 761ms | 24.93 / 0.7578 / 0.332 |
> | Summation | 30h 6min | 461ms | 25.00 / 0.7628 / 0.304 |
> | Ours | 30h 13min | 462ms | 25.26 / 0.7676 / 0.303 |
>
> Moreover, we also report the inference time and \#Flops when generating a $1792 \times 1008$ output in the following table. It shows that our method outperforms state-of-the-art ones in inference computation cost, which further illustrates the inference efficiency of our proposed method.
> | Methods | #FLOPs | Inference Time |
> |:---:|:---:|:---:|
> | FGDNet [b] | 38.67G | 476ms |
> | SANet [c] | 161.06G | 2761ms |
> | CUNet [d] | 14.48G | 542ms |
> | MNNet [e] | 23.68G | 1360ms |
> | DVN [a] | 104.05G | 761ms |
> | Ours | 13.17G | 462ms |

---

> ### Author Response · Authors · 2023-11-22
> **Response to Reviewer D7EG (2/2)**
>
> `4.` **Adaptability across diverse imaging conditions and noise levels.**
>
> It is worth noting that the proposed method is tailored to address the color and structure discrepancies between NIR-RGB images, which is an inherent issue of NIR-assisted image denoising. In other words, the issue does not disappear as imaging conditions and noise levels change. Thus, our method has the potential to be adapted to different imaging conditions and noise levels.
>
> On the one hand, the proposed NAID dataset covers three different lighting environments, i.e., 2 lux, 5 lux, and 10 lux respectively. For these lighting conditions, our extensive experiments have validated the effectiveness of our method. For lighting conditions beyond the range, it may be more appropriate to collect new data with our data acquisition pipeline and retrain our models on it.
>
> On the other hand, we have conducted extensive experiments on both synthetic DVD [a] and real-world NAID datasets, where the former covers two noise levels and the latter covers three ones. The results show that the proposed method enables performance improvement for all noise levels. And the noise level the higher, the improvement is more significant. It demonstrates that the proposed method can be effectively applied to different noise levels.
>
> `5.` **Application to other types of images.**
>
> The proposed method is elaborately designed to overcome the color and structure discrepancies between NIR-RGB images. It is simple yet efficient for real-world NIR-assisted image denoising. Simultaneously, we indeed do not know so much about other image types (e.g., medical images and satellite images) and tasks. However, when a similar content inconsistency issue is presented in other images or other tasks, we believe that our method will be also valuable.
>
> `6.` **Comparison between NAID and other benchmark datasets in diversity and size.**
>
> As far as we know, for real-world NIR-assisted denoising, Dark Flash Photography [f] is the only publicly available dataset. However, it only consists of five image pairs. It is hard to satisfy data-driven deep learning methods.
>
> In contrast, the proposed NAID dataset includes 300 NIR-RGB paired images, and covers diverse real-world scenarios and various noise levels, which has the potential to serve as a benchmark for further studies. Moreover, we promise the proposed dataset will be publicly available, which may be advantageous for future studies.
>
> Reference
>
> [a] Jin, Shuangping, et al. "DarkVisionNet: Low-light imaging via RGB-NIR fusion with deep inconsistency prior." AAAI. 2022.
>
> [b] Zehua Sheng, et al. "Frequency-domain deep guided image denoising." TMM. 2022.
>
> [c] Zehua Sheng, et al. "Structure aggregation for cross-spectral stereo image guided denoising." CVPR. 2023.
>
> [d] Xin Deng, et al. "Deep convolutional neural network for multi-modal image restoration and fusion." TPAMI. 2020.
>
> [e] Shuang Xu, et al. "A model-driven network for guided image denoising." Information Fusion. 2022.
>
> [f] Zhi, Tiancheng, et al. "Deep material-aware cross-spectral stereo matching." CVPR. 2018.

---

### Official Review · Reviewer_HWD6 · 2023-11-01

**Soundness:** 3 good
**Presentation:** 3 good
**Contribution:** 3 good
**Rating:** 8
**Confidence:** 2

**Summary:**

The authors propose a new NIR-Assisted Image Denoising dataset which covers diverse scenarios as well as various noise levels. They also propose an efficient selection fusion module in order to address the inconsistency between NIR-RGB images. This module can be plug-and-played into the existing denoising networks to merge the NIR-RGB features. The experimental results demonstrate the effectiveness of the proposed method.

**Strengths:**

1. The new dataset is valuable.
2. The proposed selective fusion module is simple yet effective.
3. The overall writing quality is good.

**Weaknesses:**

There is one place that is not clear.
The authors claim that "Therefore, we deploy a GMM to handle color discrepancy first followed by an LMM dealing with structure discrepancy". Is there any supporting evidence that the GMM handles the color discrepancy while the LMM handles the structure discrepancy?

**Questions:**

See the weakness section.

---

> ### Author Response · Authors · 2023-11-22
> **Response to Reviewer HWD6**
>
> We thank the reviewer for the valuable comments and suggestions. We appreciate the reviewer's questions and hope our responses can address the concerns.
>
> `1.` **Evidence to support that GMM handles the color discrepancy while LMM handles the structure one.**
>
> We have provided the visualization comparison results in Figure A of the appendix by removing GMM and LMM separately. It shows that GMM results in better color recovery and LMM results in better structure recovery, which supports that GMM handles the color discrepancy and the LMM handles the structure one.

---

### Official Review · Reviewer_YVkk · 2023-11-01

**Soundness:** 2 fair
**Presentation:** 3 good
**Contribution:** 3 good
**Rating:** 5
**Confidence:** 4

**Summary:**

In this paper, the authors propose an efficient lightweight selective fusion module that can be plug-and-played into any denoising network to accomplish the NIR-assisted denoising. In addition, this paper constructs a corresponding real-world NIR-assisted image denoising dataset.

**Strengths:**

Plug-and-play lightweight fusion modules can be embedded in any single-image denoising network.
Real-world dataset is a promising solution to the current lack of data for this problem and are expected to be the baseline benchmark dataset for future research.

**Weaknesses:**

1. The ablation experiment lacked a probe to see if the loss of multiple scales would affect the final output picture.
1. The lightweight fusion module in this paper does not contribute enough to the technology and simply fuses the features.
1. This paper does not demonstrate whether the improvement in denoising performance is due to the superiority of the method or only to the inclusion of NIR information.

**Questions:**

Are the methods compared in the paper all image restoration assisted by NIR images? If not, is retraining performed for the comparison?

---

> ### Author Response · Authors · 2023-11-22
> **Response to Review YVkk (1/2)**
>
> We thank the reviewer for the valuable comments and suggestions. We appreciate the reviewer's questions and hope our responses can address the concerns.
>
> `1.` **Effect of multi-scale loss.**
>
> We apologize for the lack of explanations and effect of multi-scale loss. The multi-scale loss calculates the difference between output and supervision at different scales, and it can slightly improve performance with few increasing training costs. Taking NAFNet [a] as an example, we evaluate its effect by comparing it with naive $\ell_2$ loss. The following table shows that it improves both single-image and NIR-assisted denoising performance. And the improvement is more obvious for the latter. It is worth noting that we conduct all ablation studies with the multi-scale loss, thus the comparisons between them are fair.
> | Denoising Methods | Input Images | Loss Function | PSNR / SSIM / LPIPS |
> |:---:|:---:|:---:|:---:|
> | Single-Image | RGB | Naive $\ell_2$ | 24.72 / 0.7476 / 0.343 |
> | Singel-Image | RGB | Multi-Scale  | 24.73 / 0.7483 / 0.336 |
> | NIR-Assisted | RGB and NIR | Naive $\ell_2$ | 25.20 / 0.7691 / 0.299 |
> | NIR-Assisted | RGB and NIR | Multi-Scale | 25.26 / 0.7695 / 0.299 |
>
> We will add this in the revision.
>
> `2.` **The lightweight fusion module does not contribute enough and simply fuses the features.**
>
> Here we elaborate on the proposed selective fusion module (SFM) to further show its contributions.
>
> (1) The proposed SFM is a problem-oriented design module.
>
> Color and structure discrepancies between RGB and NIR images are the main challenges of NIR-assisted denoising. SFM is elaborately designed towards the issue. Specifically, it consists of a global modulation module (GMM), a local modulation module (LMM) and a fusion operation, where GMM modulates features globally to overcome the color discrepancy and LMM does that locally to overcome the structure one.
>
> (2) The proposed SFM is a simple yet efficient module.
>
> Extensive quantitative and qualitative experiment results show that the proposed method outperforms state-of-the-art ones. Besides, we have tried some prevailing methods like feature summation, channel attention [b],  self-attention [c], and cross-attention [c] in experiments. However, as shown in the following table, they do not achieve competitive performance while introducing higher computation costs.
> | Fusion Methods | PSNR / SSIM / LPIPS | Inference Time |
> |:---:|:---:|:---:|
> | Summatioin | 25.03 / 0.7647 / 0.304 | 461ms |
> | Channel Attention | 25.12 / 0.7664 / 0.309 | 461ms |
> | Self-Attentnion | 25.12 / 0.7659 / 0.307 | 1614ms |
> | Cross-Attention | 25.18 / 0.7692 / 0.304 | 1712ms |
> | Ours | 25.26 / 0.7695 / 0.299 | 462ms |
>
> (3) The proposed SFM is a plug-and-play module.
>
> It can be seamlessly integrated into existing denoising networks with few increasing computation costs.
>
> Combining the above reasons, we finally choose the simple yet effective, plug-and-play SFM as our solution. Besides, we propose a real-world NIR-assisted image denoising (NAID) dataset that covers diverse scenarios as well as various noise levels and is expected to serve as a benchmark for future studies. More importantly, we introduce NIR-assisted image denoising from a synthetic manner into the real world.
>
> We will revise the relevant descriptions to make SFM clearer in the revision.
>
> `3.` **Is the performance improvement due to the method's superiority or only to the inclusion of NIR information?**
>
> Here we provide a detailed and clearer description of that, and we will add it in the revision.
>
> We take NAFNet as a denoising backbone and compare our method with a single-image denoising manner (only using RGB images) and a naive fusion manner (i.e., simple summation of RGB features and NIR ones). The results on NAID dataset are shown in the following table. We employ PSNR, SSIM and LPIPS as evaluation metrics.
>
> On the one hand, the inclusion of NIR information improves RGB image denoising, even simply fusing their features by summation. On the other hand, simple summation is not the optimal solution as it ignores color and structure discrepancies between NIR-RGB images. Taking this into account, the proposed SFM further improves performance, and the improvement is more significant when the noise level is higher. It illustrates the superiority of the proposed method.
> | Input Images | Fusion Methods |  Low-Level Noise | Middle-Level Noise | High-Level Noise |
> |:---:|:---:|:---:|:---:|:---:|
> | RGB | None |  25.71 / 0.7780 / 0.294 | 24.76 / 0.7482 / 0.335 | 23.71 / 0.7186 / 0.378 |
> | RGB and NIR | Summation |  25.92 / 0.7883 / 0.277 | 25.00 / 0.7682 / 0.304 | 24.16 / 0.7429 / 0.329 |
> | RGB and NIR | Our SFM | 26.06 / 0.7906 / 0.274 | 25.26 / 0.7676 / 0.303 | 24.48 / 0.7503 / 0.321 |

---

> ### Author Response · Authors · 2023-11-22
> **Response to Review YVkk (2/2)**
>
> `4.` **Whether all compared methods are assisted by NIR images? Whether they are retrained for the comparison?**
>
> In Tables 2 and 3, the methods marked as 'Single-Image Denoising' in the top three rows do not use NIR images, and the bottom eight ones marked as 'NIR-Assisted Denoising' remove noise assisted by NIR images.
>
> All compared methods are retrained with the same configurations on synthetic DVD [d] and our real-world NAID datasets respectively. Thus, the comparisons are fair.
>
> Compared to the most competitive one DVN [d],  our method achieves 0.39dB PSNR gain when $\sigma = 4$ and 0.43dB PSNR gain when $\sigma = 8$ on DVD dataset. It also achieves 0.1dB, 0.33dB, and 0.53dB improvement on PSNR in dealing with low-level, middle-level, and high-level noise on the NAID dataset, respectively.
>
> Reference
>
> [a] Chen, Liangyu, et al. "Simple baselines for image restoration." ECCV. 2022.
>
> [b] Zhang, Yulun, et al. "Image super-resolution using very deep residual channel attention networks." ECCV. 2018.
>
> [c] Liu, Ze, et al. "Swin transformer: Hierarchical vision transformer using shifted windows." ICCV. 2021.
>
> [d] Jin, Shuangping, et al. "DarkVisionNet: Low-light imaging via RGB-NIR fusion with deep inconsistency prior." AAAI. 2022.

---

### Meta-Review · Area_Chair_yTBt · 2023-12-11

**Metareview:**

This paper proposes a selective fusion approach to solve NIR-assisted image denoising and also constructs a paired NIR-assisted real-world image denoising dataset.

It received reviews with mixed ratings. The major concerns include the limited technical contributions, insufficient ablations, and unclear motivations.

Specifically, the motivation of SFM is not explained clearly. In addition, the evaluation of SFM should be better provided as mentioned by Review YVkk, Reviewer D7EG, and Reviewer KiWB.

The rebuttal partly solve the concerns of reviewers. However, the technical contributions and motivations are still not explained well.  Although the attitude of Reviewer HWD6 on the paper is positive, the review confidence rating is not high.

Based on the recommendations of reviewers, the paper is not ready for ICLR.

**Justification For Why Not Higher Score:**

N/A

**Justification For Why Not Lower Score:**

N/A

---

### Decision · Program_Chairs · 2024-01-16

Reject